# Time-resolved, integrated analysis of clonally evolving genomes

Carine Legrand[1,2]*, Ranja Andriantsoa[1], Peter Lichter[3,4], Günter Raddatz[1], Frank Lyko[1]

**1** Division of Epigenetics, DKFZ-ZMBH Alliance, German Cancer Research Center, Heidelberg, Germany,
**2** Université Paris Cité, Génomes, biologie cellulaire et thérapeutique U944, INSERM, CNRS, Paris, France,
**3** Division of Molecular Genetics, German Cancer Research Consortium (DKTK), German Cancer Research Center (DKFZ), Heidelberg, Germany, **4** Molecular Precision Oncology, National Center for Tumor Diseases, Heidelberg, Germany

* carine.legrand@inserm.fr

## Abstract

Clonal genome evolution is a key feature of asexually reproducing species and human cancer development. While many studies have described the landscapes of clonal genome evolution in cancer, few determine the underlying evolutionary parameters from molecular data, and even fewer integrate theory with data. We derived theoretical results linking mutation rate, time, expansion dynamics, and biological/clinical parameters. Subsequently, we inferred time-resolved estimates of evolutionary parameters from mutation accumulation, mutational signatures and selection. We then applied this framework to predict the time of speciation of the marbled crayfish, an enigmatic, globally invasive parthenogenetic freshwater crayfish. The results predict that speciation occurred between 1986 and 1990, which is consistent with biological records. We also used our framework to analyze whole-genome sequencing datasets from primary and relapsed glioblastoma, an aggressive brain tumor. The results identified evolutionary subgroups and showed that tumor cell survival could be inferred from genomic data that was generated during the resection of the primary tumor. In conclusion, our framework allowed a time-resolved, integrated analysis of key parameters in clonally evolving genomes, and provided novel insights into the evolutionary age of marbled crayfish and the progression of glioblastoma.

## Author summary

Genomes evolve under the accumulation of mutations, and under the pressure of selective forces. While additional mechanisms are at play in sexually reproducing species, this is not the case in clonal genomes. Our study focuses on a parthogenetic animal and on cancer, since both possess a clonal genome, and in both cases evolutionary forces are key to understand expansion. We used modelling of mutation accumulation, in combination with Darwinian selection and with clock-like mutagenic processes. Using this framework, we showed a remarkably recent emergence date for *P. virginalis* and established its potential as a model system for clonal genome evolution. We highlighted subtle temporal

**Data Availability Statement:** Sequence data for marbled crayfish data have been deposited as a National Center for Biotechnology Information BioProject (accession number: PRJNA356499). Glioblastoma data were accessed from the

European Genome-phenome Archive (EGA) database, with accession number: EGAS00001003184 (glioblastoma).

**Funding:** ANR STEM R20117HH

dynamics of selection in tumor samples, and showed that tumor cell survival was correlated with the time to recurrence. Our findings illustrate the potential of this framework for modelling of clonal evolution and for the use of evolutionary parameters in a clinical context.

## Introduction

The evolution of genomes is shaped by many factors, among which the random accumulation of mutations over time plays a fundamental role [1,2]. Because of this, the characteristics of mutated sites can be used as a lens to observe the evolutionary processes that shaped the genome in the past. For instance, the ratio of nonsynonymous to synonymous mutations reveals if selection had an impact on the genome [3]. Furthermore, biological processes, including clock-like processes, leave a footprint in the form of recently developed mutational signatures [4]. The frequency of mutated alleles could also elucidate the timeline of the evolving genome, or selection [5], but this can be confounded by stochastic drift, or by alterations of ploidy [6,7]. These individual evolutionary parameters may or may not exert an influence on each other. An integrated analysis aims at modelling these elements and their interplay, in order to gain a better understanding on their role on the genome.

Far from being homogeneous, the probability of a mutation depends on many factors such as the genomic location [3], mutator alleles, local nucleotide context or mutagenic exposures [4]. Other genomic modifications include recombination in sexual reproduction, copy number variants and genomic rearrangements, gene transfers and hybridization. The capacity of any genomic modification to be inherited is partly stochastic, for instance through genetic drift [8], but can be favored or disfavored by positive or negative selection. Genome evolution was historically observed through the analysis of phenotypes [9], and can now be determined more precisely using high-throughput sequencing in parallel with experimental or cohort settings, such as mutation accumulation experiments, or the analysis of genetic trios [10,11].

Under certain conditions, genomes can evolve clonally, which involves a more limited set of mechanisms. This is particularly relevant for asexually reproducing species and for human cancers. Mutation rate, growth and variant frequencies are key parameters of clonally evolving genomes [12]. They determine the speed of evolution and function under the influence of selective pressures.

Selection can be quantified using the ratio of nonsynonymous to synonymous mutations (dNdS), where a lower than expected ratio indicates purifying (negative) selection, and a higher ratio indicates positive selection. The expected dNdS ratio is not trivial to determine, because the identity of the neighboring genomic bases, or the location of the mutation in the gene transcript, can alter the frequency of certain mutations. Selection is also a multifaceted, dynamic event which actively depends on the environment [13,14]. Of note, the notion of stochastic drift, which corresponds to the random variation of the frequency of alleles (or, of clones), is a process distinct from selection. Stochastic drift can happen without the advent of selection (neutral drift), or in addition to it [1].

A prominent species with a clonally evolving genome is the marbled crayfish (*Procambarus virginalis*), a newly discovered freshwater crayfish [15,16]. Marbled crayfish reproduce by apomictic parthenogenesis, with the offspring being genetically identical copies of their mothers [17,18]. Interestingly, genetic analyses have suggested that the global marbled crayfish population represents a single clone, indicating that it was formed only recently and by a single foundational animal [19,20]. Morphological and genetic examinations have identified *Procambarus*

*fallax*, a sexually reproducing slough crayfish from Florida, as the parent species of the marbled crayfish [21]. Furthermore, a recent phylo-geographic analysis of *P. fallax* suggested that the anthropogenic transport and cultivation of a triploid and parthenogenetically reproducing *P. fallax* specimen could be the origin of the marbled crayfish [21]. The offspring of this foundational specimen were subsequently distributed through the aquarium trade and released into various environments, thus forming numerous stable wild populations around the globe [20]. However, important details about the speciation of the marbled crayfish are not known and need to be supported by genetic analysis.

Clonal genome evolution also plays an important role in cancer formation. Indeed, cancer genome evolution is characterized by the accumulation of somatic mutations into a pathogenic tumoral genome. Several authors have described the critical role of mutational patterns and selection in cancer [1,3,22], while neutral evolution is still debated [5,7]. In glioblastoma, the analysis of tumor trajectories revealed a tumor initiation years before diagnosis [23]. Consequently, it would be of great interest to infer evolutionary parameters over the course of tumor progression.

In this study, we aimed to develop an integrated analysis of clonal genome evolution. To this end, we reformulated the dependence of mutation accumulation on variant allele frequency, and used this formulation to determine the links between the mutation rate, growth and survival rates. We further integrated these parameters with selection estimates, obtained from the non-synonymous to synonymous ratio. Finally, we integrated time estimates in our model, based on clock-like mutational signatures. We applied our approach to the clonally evolving marbled crayfish. We provided a detailed view of mutation accumulation and selection, and estimated the time of speciation. We further applied our framework to clonal genome evolution in cancer, using recently published samples of primary and recurrent glioblastoma [23].

## Results

The genetic near-monoclonality of the marbled crayfish population [19,20] establishes this species as an excellent model system for studying clonal genome evolution. In order to assess the mutation rate of the *P. virginalis* genome, we used paired-end whole-genome sequencing at an average of 17x coverage (per strand), for a line of one ancestral animal and two direct descendants from our laboratory colony of *P. virginalis*, that were sampled over a period of seven years (Fig 1A). The mutation rate was calculated as the average number of de novo mutations in animals 34 and 35 as compared to animal 1, per nucleotide and per year. From these samples, we obtained a range for the mutation rate equal to
$\mu = [3.51 \cdot 10^{-8}; 1.17 \cdot 10^{-4}]/nt/y$, The lower bound corresponds to strictly filtered mutations divided by the number of sites in the whole genome, while the upper bound corresponds to the number of mutations remaining after a relaxed filtering, divided by the number of callable sites. The range of the mutation rate of *P. virginalis* genome overlaps with known mutation rates in arthropods, and is also comparable to the mutation rates observed in human somatic healthy or cancerous cells (Fig 1B).

We made an evaluation of the evolutionary age of *P. virginalis*, using a Markov Chain Monte Carlo with Bayesian evolutionary analysis [35] on whole-genome sequencing datasets from 13 animals (Fig 1C). We generated 10 million states, which allowed convergence of the sampled states, and led to adequately large effective sample sizes (see Methods for details). The resulting coalescent tree showed that animals 1, 34 and 35 correctly clustered together, as well as animals from German wild populations (Hannover, Reilingen, Moosweiher) and from the likely foundational laboratory lineage of the German wild populations (Heidelberg).

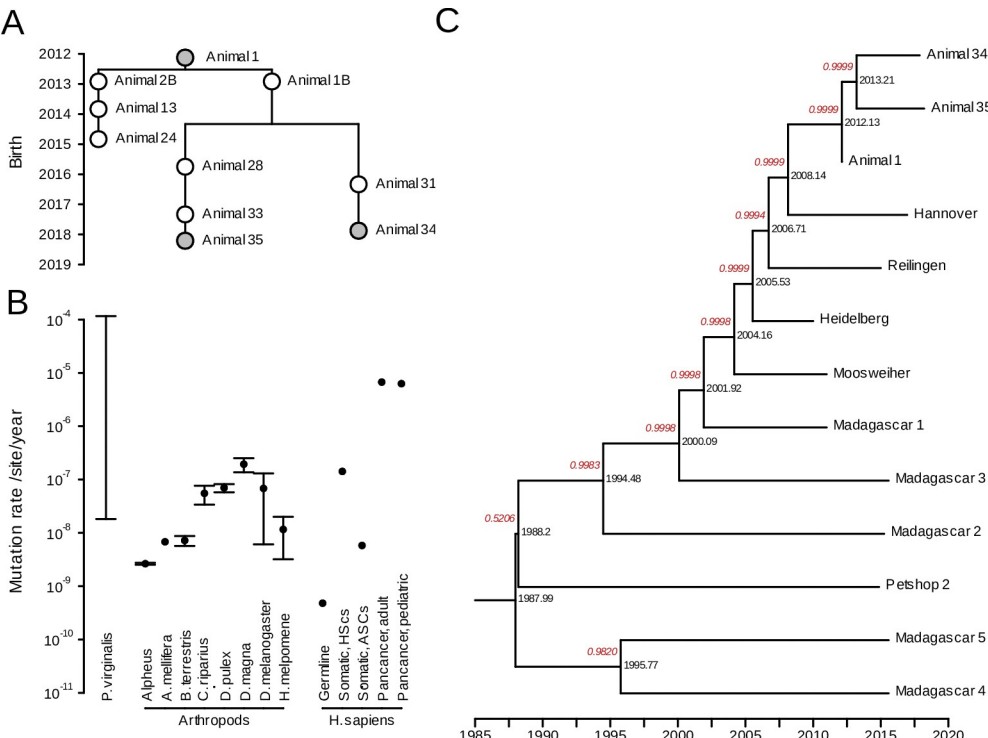

**Fig 1. Mutation rate of *P. virginalis* and coalescent.** (A) Genealogy of laboratory animals, with sequenced animals marked in grey. (B) Mutation rate in *P. virginalis*, in other arthropods (Silliman et al. 2021 [24], Yang et al. 2015 [25], Liu et al. 2017 [26], Oppold et al. 2016 [27], Flynn et al. 2017 [13], Ho et al. 2020 [28], Keightley et al. 2014 [29], Keightley et al. 2015 [30]), and in *Homo sapiens* (Ohno et al. 2019 [31], Lee-Six et al. 2018 [32], Blokzijl et al. 2016 [33], Martincorena et Campbell 2015 [10], Ma et al. 2018 [34]). HSC: Hematopoietic Stem Cells, ASCs: Adult Stem Cells (small intestine, colon and liver). Error bars correspond to 95% confidence intervals. (C) Coalescent tree based on a constant mutation rate and sequences of sampled animals. The posterior probability of each branch is indicated in red.

Furthermore, samples from Madagascar formed a separate branch. Interestingly, Petshop 2 [19] was nested in the branch of animals from Madagascar. This is consistent with the notion that the Malagasy population was founded by an animal that was originally obtained from a German pet shop. Posterior probabilities (Fig 1C, red annotations) indicate highly probable branching for all but the top coalescent event, which has 0.5206 probability. From this tree, the most recent common ancestor of the 13 animals occured in 1988.0 (95% CI: [1986.1; 1989.8]). This is anterior, and therefore broadly consistent, with the first documented appearance of *P. virginalis* in 1995 [16].

We next modeled mutation accumulation under a fast growth scenario. In this model, the number of mutations *dM*, arising in a time increment, scaled with the mutation rate and other evolutionary parameters (S1 Text, p.2, expression (1)). Then, we noticed that allele frequency could be expressed based on ploidy and the number of animals [5], and that the number of animals could be further expressed using the rate of reproduction, offspring survival, and the population size (S1 Text, p.4, expression (7)). This reformulation of allele frequency appeared advantageous, because it could then be used to simplify the expression of *dM*. As a result, under this fast growth scenario, *dM* could be simply expressed using the mutation rate, constant terms, and allele frequency (S1 Text, p.5, expression (12)). As a consequence, mutation accumulation gave information on the mutation rate, but not on selection (S1 Text). This provided the rationale for examining the dynamics of the mutation rate in *P. virginalis* using the *M(1/f)* curve, where *M* is the number of mutations and *f* is the allelic frequency (Fig 2A). The

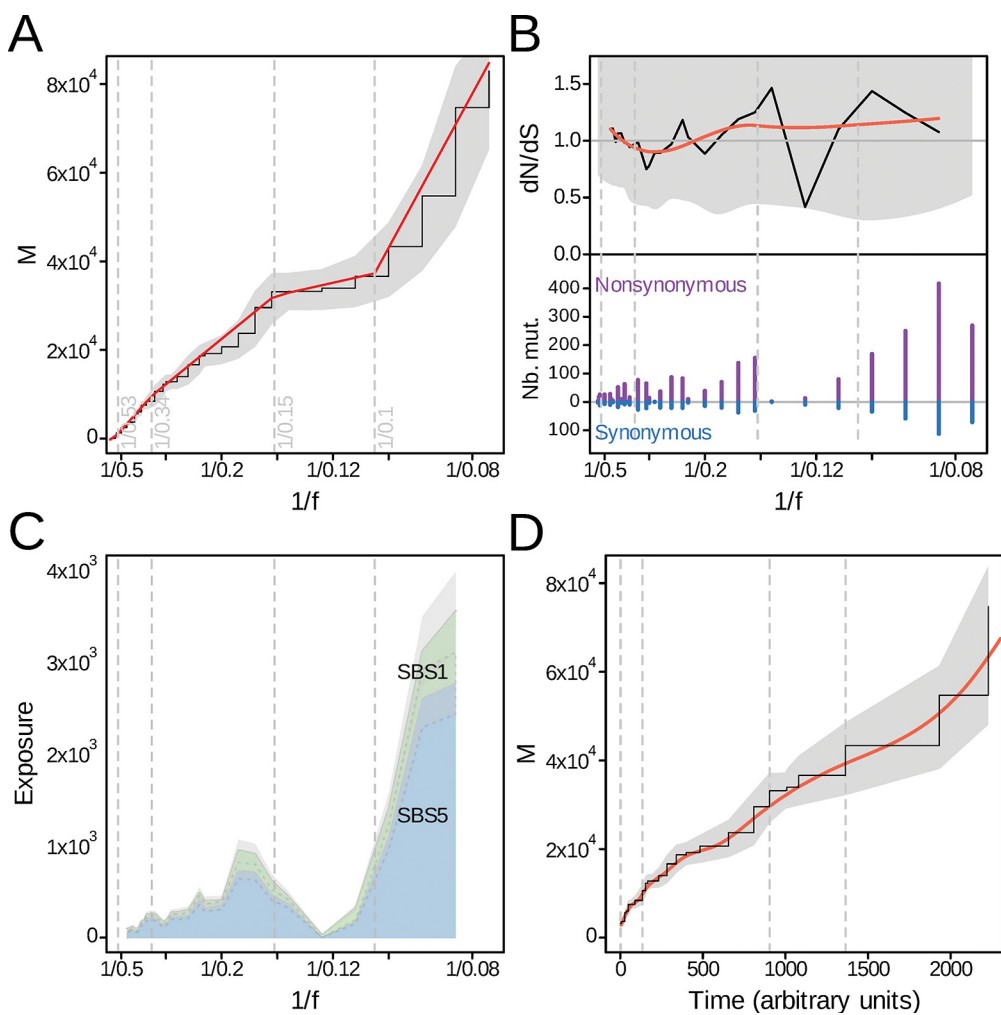

**Fig 2. Mutation accumulation, selection and time course of *P. virginalis* genome evolution.** (A) Mutation accumulation as a function of the inverse allele frequency 1/f (black) and phases from automated segmentation (breakpoints in grey, segments in red). The confidence band at 95% level is shown in grey. (B) Non-synonymous to synonymous ratio (*dNdS*). The smoothed ratio is shown in red. (C) Stack plot of exposure, the contribution of each mutational signature. This includes clock-like single-base substition 1 (SBS1) signature, and clock-like single-base substitution 5 (SBS5) signature. Confidence bands at 95% probability are indicated in grey. (D) Mutation accumulation as a function of time. The confidence band at 95% is shown in grey and the smoothened mutation accumulation is shown in red.

resulting curve suggested that the mutation rate changed over time, with 4 phases delineated by a segmented regression (Fig 2A; *p* = 0.06). The mutation rate was reduced in phase 3, as compared to phases 1 and 2, and increased in phase 4 (Fig 2A). Under our model, selection *s* is not observable using *M(1/f)* (S1 Text, Eq 12). We therefore used the ratio of non-synonymous to synonymous mutations to estimate *s* (Fig 2B). The resulting values were compatible with unity, suggesting the absence of selection.

In order to obtain time-resolved estimates, we then used previously established clock-like mutational single-base signatures (SBS1 and SBS5) [36–38] as a proxy for the time course of mutation accumulation (Fig 2C). Because mutational signatures are currently lacking in arthropods, but the underlying mechanisms appear conserved in evolution [39,40], we used human mutational signatures. We further assumed that the arrow of time from past to present

corresponds to the arrow of increasing *1/f*. To obtain a time course in arbitrary units, we calculated the integral of the clock-like components of mutation accumulation (Fig 2D, Methods). According to the mathematical model, the slope of this curve is proportional to the mutation rate as a function of time (S1 Text, Eq 12). The results (Fig 2D) showed that this mutation rate exhibited less variation than the mutation rate per division (Fig 2A). Because the temporal and per-division mutation rates differ in particular by the growth rate (S1 Text), this might indicate fluctuations in the growth rate of the marbled crayfish population. As a whole, our analyses suggested distinct phases, detected significant variations of evolutionary parameters in *P. virginalis*, and allowed to trace its speciation to a time point that is consistent with biological records.

In *P. virginalis*, we developed a framework to analyse the evolution of a clonal genome, which is driven by germline mutations. This framework can in principle also be applied to analyse the clonal evolution of a tumor genome, which is driven by somatic mutations. Since glioblastoma is a high grade tumor with systematic recurrence and poor patient survival, a better understanding of its evolutionary parameters is important. We therefore applied our framework to a published set of whole-genome sequencing data of primary and recurrent glioblastoma tumors [23]. This study also estimated the age of primary tumors, allowing further data integration. Based on the curve *M(1/f)*, we generated mutation rate profiles (Figs 3A, see S1A for individual samples), which we further segmented into phases (Fig 3A, p < 2.2x10⁻¹⁶). The results indicated distinct variations in the mutation rate in primary and recurrent samples (Fig 3; S1 Table). In the exemplary sample 1 in Fig 3A, the segmentation separates 5 phases significantly. K-fold cross-validation showed that the mean square error was 2379.4 for 4 phases and 1184.0 for 5 phases. The difference between test and validation mean square errors was 291.0 for 4 phases and 135.5 for 5 phases. These results strongly suggest the absence of overfitting. After we excluded the outermost phases 1 and 5, where changes in mutation frequencies may correspond to an early slow growth phase, or where our analysis may miss low-frequency mutations [23], the mutation rate per division decreased steadily in phases 2–4.

We next looked at selection using the *dN/dS* ratio. Taking confidence bounds into account, the results were compatible with neutral selection for most tumors (Figs 3B, S1B per sample). However, 11 primary tumor samples showed evidence of negative selection during intervals, for instance sample 35 (S1B Fig for sample 35). We also observed evidence for positive selection in two primary tumor samples (Samples 2 and 7, S1B Fig). Interestingly, 7 out of 9 recurrent tumor samples underwent prolonged phases of negative selection (for example, sample 4, 1/f in [1/0.5; 1/0.1], S1B Fig), while 2 samples exhibited short phases of negative selection. No recurrent tumor sample showed any significant phase of positive selection.

We next determined the timeline of tumor evolution, by examining the frequency of the stable, clock-like SBS1 signature. Using the information on the clock-like signature SBS1, and using Eq (9) (Methods), we reconstructed *M* as a function of time (Figs 3D, S1D per sample), in arbitrary units. The slope of this curve is proportional to the mutation rate per time unit. Similar to the mutation rate per division in Fig 3A, the mutation rate per time unit decreased from phase 2 to phase 4, though less markedly. Furthermore, similar to the situation for *P. virginalis*, the difference observed in sample 1 might indicate fluctuating growth during phases 2–4. More specifically, differences between per-division and temporal mutation rates corresponded in our model to the growth rate, the survival rate, and the number of cells (S1 Text, Eq 8).

We next asked if these terms could be used to characterize the set of 42 primary and recurrent tumor pairs. Using temporal and per division mutation rates, we could reconstruct these terms, which correspond to the product $\omega\gamma N$. This quantity corresponds to growth ($\omega$), modulated by the survival rate ($\gamma$), and scaled by the number of cells ($N$). In other terms, the product

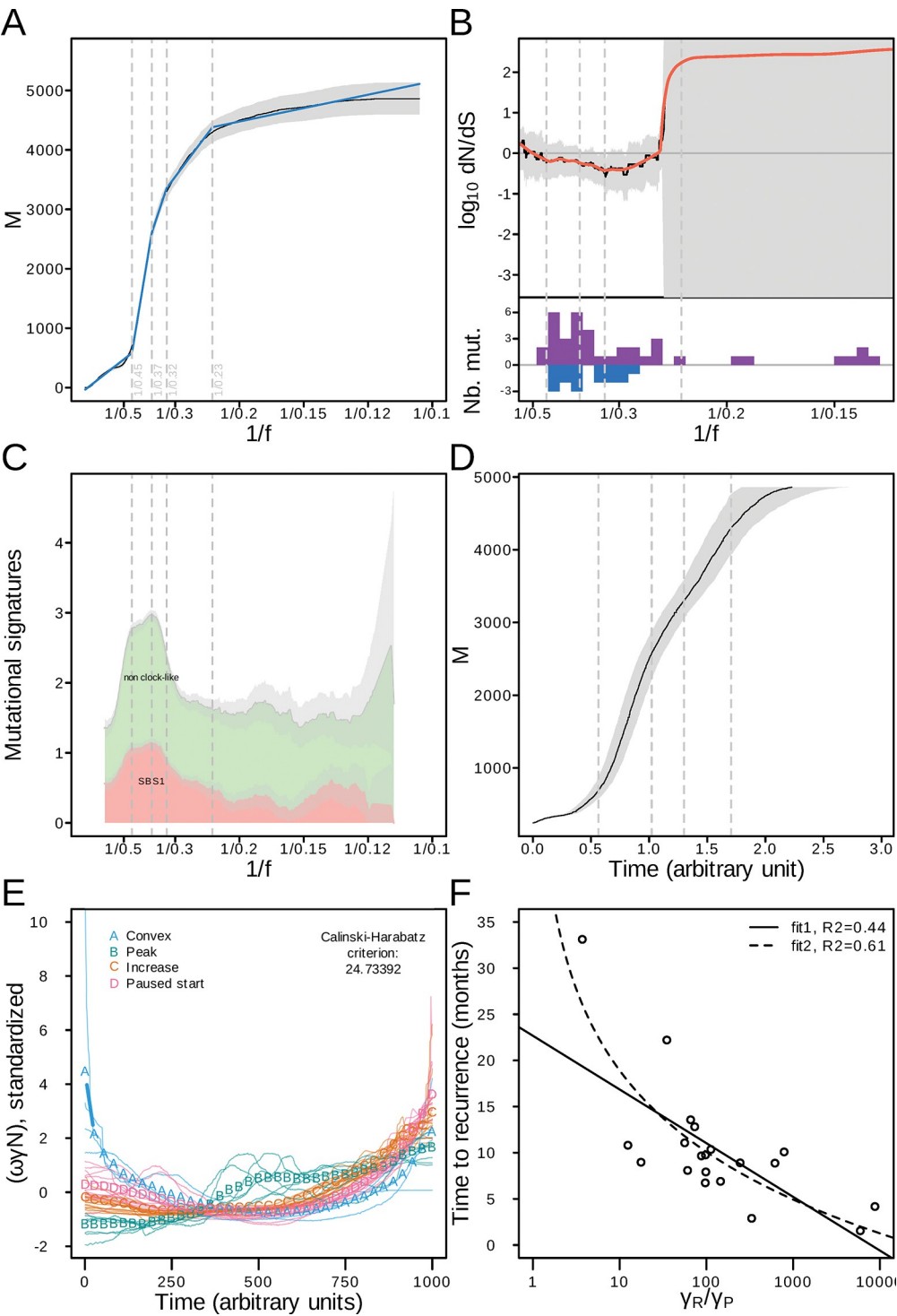

**Fig 3. Mutation accumulation, selection and time dynamics of a representative glioblastoma tumor (patient 1, primary tumor), expansion patterns and survival ratio.** (A) Mutation accumulation as a function of the inverse allele frequency *1/f* (black) and phases from automated segmentation (breakpoints are indicated as dashed vertical lines, segments are indicated in blue). The confidence band at 95% level is shown in grey. (B) Non-synonymous to synonymous ratio *dNdS*. In the lower inset, purple and blue bars show non-synonymous and synonymous mutations, respectively. The smoothened ratio is shown in red. (C) Clock-like and non-clock-like mutational signatures. (D) Mutation accumulation as a function of time. (E) Stratification of expansion curves $\omega\gamma N$ into 4 subgroups: A: Convex, B: Peak, C: Increase, D: Paused Start (42 primary tumor GBM samples). (F) Dependence of time to recurrence on the

$\gamma_R/\gamma_P$ ratio. Fit1 corresponds to a linear regression of time versus $\log_{10}(\gamma_R/\gamma_P)$, with intercept = 19.511 (standard error SE = 2.544) and slope = -5.819 (SE = 1.455), fit2 corresponds to a linear regression of time versus $\log_{10}(\log_{10}(\gamma_R/\gamma_P))$, with intercept = 18.922 (SE = 1.806) and slope = -29.321 (SE = 5.285).

$\omega\gamma N$ reflects the effective expansion of the tumor. Consequently, we denoted the product $\omega\gamma N$ as expansion parameters in the following. We examined the corresponding curves and found that unsupervised clustering allowed us to classify the tumors into four subgroups: (A) Convex, (B) Peak, (C) Increase and (D) Paused Start (Fig 3E).We then looked at a possible association between the patterns of the $\omega\gamma N$ curve in the primary tumors, and the time to the recurrence, but the results were inconclusive ($p = 0.4916$, n = 19). However, the time difference between the resection of the primary tumor and the resection of the recurrence is known for a subset of samples, and the age of tumors was estimated previously [23]. This allowed us to transform the time course from arbitrary units into real units (Methods, Eq 2, S2 Fig). Furthermore, we extended our modelling to be able to express the transition from the primary to the recurrent tumor (Methods, Eq 4). With this, we could determine the tumor survival ratio from time estimates. Using the previously established values of 2 years and 7 years [23] as the lowest and highest limits for the time course of the primary tumors, we could determine a range for the value of the tumor survival ratio $\gamma_R/\gamma_P$ for each individual sample (Methods, Eq 8, S2 Table). As a result, the lowest value of the ratio $\gamma_R/\gamma_P$, corresponding to a tumor emergence about 2 years before diagnosis, was always higher than 1 (median = 27.8 [17.4; 54.0] for the lower bound, median = 97.5 [60.9; 189.0] for the upper bound, n = 20 samples). These results indicated that tumor cell survival was higher at the start of the recurrence than at the end of the primary tumor growth. Not surprisingly, $\gamma_R/\gamma_P$ ratios were associated with the time to recurrence (Fig 3F, $p_{adj} = 1.258\times10^{-3}$ and $p_{adj} = 8.649\times10^{-4}$), with higher $\gamma_R/\gamma_P$ ratios corresponding to shorter time to recurrence. Collectively, these results uncover substantial variations of evolutionary parameters among glioblastoma samples, and provide an improved understanding of growth and survival in tumor subgroups.

## Discussion

In this study, we presented an integrated framework to analyse the evolution of clonally evolving genomes. We first determined the mutation rate of *P. virginalis* to be in the range of [$3.51\times10^{-8}$; $1.165\times10^{-4}$]/nt/y, which encompasses the mutation rates in other arthropods, the human germline, and in human somatic healthy cells. The upper end is also comparable to microsatellites in arthropods and other species [41–44], and close to the somatic mutation rate in human cancer. Data about mutation rates in triploid genomes are scarce, and it appears possible that it may be associated with higher mutation rates. Interestingly, a high mutation rate was reported in polyploid plants ($10^{-5}$ order of magnitude [45]). We detected separate evolutionary phases, during which the mutation rate varied significantly. However, the dNdS ratio remained relatively constant, indicating the absence of selection. These findings support the argument that the mutation rate should not be considered constant [1,46,47].

We traced the speciation of *P. virginalis* to 1988 (95% confidence limits: [1986; 1990]), in agreement with first reports of this animal in 1995 [16]. This exceptionally young evolutionary age is consistent with the largely monoclonal population structure showing only incipient genetic differentiation [19,20]. It also provides experimental support for the hypothesis that the global marbled crayfish population descended from a single anthropogenic transport and release event [16,21] and further establishes the species as a unique model system.

In tumor samples, our approach allowed a single patient-level analysis of evolutionary parameters, and similarly revealed the presence of different phases, variations of the mutation

rate, and a few significant events of selection. Multisector and single-cell sequencing studies have highlighted high levels of heterogeneity and either clonal selection, or an almost complete overlap, between primary and relapsed glioblastoma tumors [48–50]. In this context, our study identified varied patterns, either of selection, or of neutral evolution. This appears comparable to previously published results [3,51], where either selection or neutrality was observed, depending on the context. Interestingly, negative selection occurred most often early in the mutation accumulation process of primary tumors, and corresponded to a low mutational load (S1 Fig, exemplary tumors 2,28,42), in agreement with recent results [52], whereas selection in recurrent tumors followed this pattern only to some extent.

Utilizing the difference between temporal and per-division mutation rate, we could stratify the samples into 4 subgroups. While clinical subtypes for GBM have been described, single-cell studies revealed high intratumoral heterogeneity [48]. In this context, our approach offers a possible alternative, although association with clinical outcome remains to be established. Building on previously estimated tumor age, we could also derive the survival ratio for tumor cells in the recurrence, relatively to the primary tumor. We found that tumor cells survive better at the start of the recurrence, albeit with important variations. This supports the notion that GBM regrowth can be more aggressive after surgical resection [53,54], possibly because resection-induced astrocyte injury can support faster growth [54], or because the tumor microenvironment can promote tumor regrowth [55–57]. Conversely, a stronger immune response might also inhibit tumor regrowth.

As our study aims to explore novel connections between diverse fields of research, we find it important to explain several limitations. For example, a more precise determination of the *P. virginalis* mutation rate could be achieved by the development of novel tools that are more amenable to triploid genomes and by experimental validation [25,29]. Also, the coalescent tree could be refined by the use of sequencing datasets with higher genome coverage to reduce the potential impact of noisy variants. For the tumor samples, it would be important to better understand the potential effect of ploidy changes. This could be achieved by restricting the analysis to diploid regions, or by the integration of ploidy information into the model. In an extended model, copy number information could also provide information on the timing of certain mutations [58,59].

In conclusion, this integrated analysis of mutation accumulation, dNdS ratio and mutational signatures provided a detailed landscape of evolutionary parameters in two paradigms of clonal genome evolution. We showed an exceptionally recent emergence date for *P. virginalis* and established its potential as a model system. We highlighted subtle temporal dynamics of selection in tumor samples, and showed that a quantification of tumor cell survival was correlated with the time to recurrence. Our findings illustrate the potential of this framework for modelling of clonal evolution and for the use of evolutionary parameters in a clinical context.

## Materials and methods

### Ethics statement

The commitee responsible for the usage of human subject data from the EGAS00001003184 study is the DKFZ-HIPO Data Access Committee of Heidelberg Center for Personalized Oncology. The approval was granted by this committee. We used the data in compliance with the declared, and approved, usage.

### *Procambarus virginalis* samples

Freshwater crayfish samples from [19] were used. Additionally, samples from Madagascar 1 sample and Moosweiher sample were resequenced (S3 and S4 Tables). Animal 1 corresponds

to the lab strain, acquired from a pet shop. New genomic DNA samples were taken from animal 34 and animal 35, which, as animal 1, also correspond to lab strains animals, and which are direct offsprings of animal 1. These new samples were prepared and submitted for whole genome sequencing following the protocol already described [19]. The genealogy and birth date of animals were retrieved from laboratory records and field records (S3 Table). Sequence data was trimmed using Trimmomatic v0.32 (settings: LEADING:3 TRAILING:3 SLIDINGWIN-DOW:4:20 MINLEN:40, adapter sequence: TruSeq3-PE). Next, trimmed data was mapped to Pvir genome assembly v04 (https://www.ncbi.nlm.nih.gov, Bioproject Accession: PRJNA356499), using Bowtie2 (v2.2.6, setting:--sensitive). The quality of this assembly is comparable to other published genomes in non-standard organisms, but there is still a higher level of fragmentation. This might preclude or render mutation detection more difficult, in these parts of the genome which are located in or at the boundary of a gap. Aligned reads were sorted, cleared from duplicates, sorted and indexed using samtools. Subsequently, variant calling was performed using Free-bayes v0.9.21-g7dd41db (parameters:--report-all-haplotype-alleles -P 0.7 -p 3--min-mapping-quality 30--min-base-quality 20--min-coverage 6--report-genotype-likelihood-max).

## Glioblastoma Multiforme samples

The glioblastoma primary and recurrent tumor samples correspond to the WGS cohort already described in [23]. In particular, summary information can be found in supplementary table 1 of [23]. After approval of the research project, access to the SNP data of primary and recurrent tumor samples, as well as time to recurrence when available, was granted.

## Mutation rate of *P. virginalis*

Mutation accumulation between animals 1 and descendant animals 34 and 35 was used to calculate the mutation rate. SNP variants were examined in terms of quality and coverage. Variants with quality≥35, coverage≥50 (strict cutoff for the lower bound of the mutation rate) and 25 (relaxed cutoff for the upper bound of the mutation rate) and ≤200 were retained for the main estimate of the mutation rate. Coverages 200 and higher exhibited altered SNP distribution and were thus excluded because possibly corresponding to a distinct part of *P. virginalis* genome (possibly highly repetitive and variable domains). The number of callable sites was 486234 for the upper bound of the mutation rate. Eligible sites were considered when the alternate allele was present in only one line [60]. Subsequently, the mutation rate per nucleotide per year was calculated as the count of biallelic mutated nucleotides in animal 34 (respectively, animal 35) as compared to animal 1, divided by the count of nucleotides in the triploid genome of *P. virginalis*, N = $10.5 \times 10^9$ (for the lower bound of the mutation rate), and by the number of callable sites (for the upper bound), and divided by the time (5.75 and 6.08 years), between the birth dates of animal 1 and 34 (respectively, birth date of animal 35). We assumed that counts of new mutations follow a Binomial distribution, and with this we determined the standard deviation on the count of mutations observed (genotyping uncertainty). Second, we assumed that the standard deviation for the date of animal birth equated to a third of the total uncertainty on time of birth. Third, we calculated the standard deviation between mutation rates for animal 34 and animal 35 (biological variability). Finally, we took the total standard deviation as the quadratic sum of these three components (assuming that the different sources of variability follow a normal distribution).

## Coalescent time

Time to most recent common ancestor for *P. virginalis* samples was determined using Bayesian evolutionary analysis by sampling trees (BEAST v1.10.4 [35]). Mutation data with quality

>35 and coverage depth >15 was used in this analysis (a coverage cutoff of 25 was not justified here because samples other than animals 1, 34 and 35 possessed a notably lower average sequencing depth). Samples birth dates were used as tip dates. Further BEAST parameters used were: simple substitution model with estimated base frequencies, strict clock, skyride coalescent prior. The length of chain for the Markov chain Monte Carlo was 10M. These parameters were built into the BEAST input file, using the utility BEAUTi. The outputs were analyzed using the utility TRACER. In particular, convergence was read from the sampled states curves of the different parameters, and effective sample sizes were adequately >100 (2984 or more), indicating sufficiently decorrelated sampled states.

## Study of mutation accumulation

An infinitesimal increment of mutations *dM* was defined from the evolutionary parameters: mutation rate, ploidy, cell survival, growth and number of cells (see S1 Text for a detailed description). Noticing that these parameters may be heterogeneous in the population (of cells, or animals), we stratified this expression for each subclone, with homogeneous parameters inside of a subpopulation (subclone). We have then noticed that observable allele frequency of a mutation is the one obtained after sequencing and SNP calling, and adapted the expression given in [5] consequently. We linked the observable allele frequency with the features of the subclone where this mutation appeared, namely the change in subclone size, the number of cells, ploidy and time (S1 Text, Eq (5)). Then, hoping to obtain an expression of allele frequency which could be instrumental in the expression of *dM*, we have determined a formula for the increment of the number of cells *dN* and for the increment of inverse allele frequency *d(1/f)*. For this latter increment, we have made the assumption that ploidy was constant. With this, we could indeed use the expression of allele frequency in the expression of *dM*. As a result; we deduced the mutation accumulation *dM* as a function of inverse allele frequency *1/f*, the mutation rate, and constants, in each subclone. Finally, the equation for mutation accumulation over all subclones *dM* was obtained by summing the individual contributions *dM* of each subclone. The mutation accumulation curve *M(1/f)*, of which slope corresponds to *dM/d(1/f)*, was plotted from SNP data (filtered by quality phred score QUAL $\geq$30, depth $\geq$10, depth of alternate allele $\geq$3), and from the corresponding allele frequencies. For uncertainties, we used bins of +/-0.25 over 1/f. In each bin, the mutation count was subjected to bootstrap resampling, which yielded a bootstrap distribution. The confidence interval at 95% was taken as the interval bounded by the 2.5% and 97.5% quantiles of the bootstrap distribution. For *P. virginalis*, the confidence interval for mutation accumulation was calculated assuming a student distribution.

## Mutation annotation and dNdS ratio

Mutations were annotated as synonymous or non-synonymous (including splice or stopgain mutations) using SNPdat v1.0.5. The bias-correcting method dNdScv [3] was tested, but could not be applied on the relatively low number of mutations at hand. Also, dNdScv does not provide longitudinal estimates. As a more pliant, but non bias-correcting method, we calculated the *dNdS* ratio as the quotient of non-synonymous mutations by synonymous mutations in a sample, divided by the average quotient in the full genome. The average quotient of non-synonymous to synonymous in humans was calculated from the reference coding sequences (hg19): fasta sequences were filtered out of coding sequences not starting with AUG or not ending with a stop codon, converted to codons, sorted and counted using a custom bash script. Using a spreadsheet, all non-redundant mutations were evaluated as synonymous or nonsynonymous. The total count of synonymous (respectively, nonsynonymous) possible mutations

per codon was taken as the count of this codon type multiplied by the number of possible synonymous (nonsynonymous) mutations evaluated from the spreadsheet. The dN/dS ratio was taken as the sum over all codons of the count of nonsynonymous mutations, divided by the sum over all codons of the count of synonymous mutations. Uncertainties were determined by bootstrap resampling of mutations, and calculations of the dNdS ratio for each bootstrap sample.

## Mutational signatures

Mutational signatures are combinations of mutations which are representative of the action of different mutagenic processes, such as exogenous (ultraviolet light) and endogenous (5-methylcytosine deamination) mechanisms, enzymatic DNA editing, DNA repair mistakes, and DNA replication infidelity [4]. The single-base substitution mutational signatures (in 3-nucleotides context) for human subjects were downloaded from the COSMIC database (https://cancer.sanger.ac.uk/signatures/; version 3.1 as of 11.08.2020). Since mutational mechanisms are conserved across the animal kingdom, we hypothesized that human mutational signatures could be applied to the marbled crayfish. For the longitudinal analysis of the mutational signatures, mutation data was binned using a bin half-width equal to 0.5 on the inverse allele frequency. The exposure of binned data was determined using R 3.5.2 with package YAPSA (version 1.8.0), where exposure corresponds to the individual contribution of each signature. Uncertainty on mutational signatures was determined by bootstrap resampling of mutations and generation of the binned data and YAPSA exposures on the resampled data. We have used 1000 bootstrap replicates as a compromise between an ideally larger (1M) number of replicates, and reasonable computing time. Large mutation sets (>100,000 mutations) were subsampled to 50,000–60,000 mutations for the bootstrap analysis. Mean, median, percentiles and 95% confidence bounds were determined using the resulting bootstrap distribution.

## Time course

We made the assumption that clock-like mutational signatures SBS1 and SBS5 were a surrogate indicator for time (SBS1 only for glioblastoma, in agreement with [36]). Because mutagenic mechanisms are conserved across the animal kingdom, and because mutational signatures in arthropods are currently lacking, we have assumed that human mutational signatures are sufficiently representative in the marbled crayfish. We further assumed that the arrow of time, could be identified with the arrow of inverse allele frequency $1/f$. Under these assumptions, an increment of time can be determined in arbitrary units, by integrating $\theta$, which denotes the exposure to clock-like mutations SBS1 or SBS5, over an increment of inverse allele frequency $1/f$. Since the exposure $\theta$ is also proportional to the number of cells in the tumor, it is necessary to normalize $\theta$, by dividing its value by the number of cells $N$. Since $N$ is proportional to $1/f$ under some assumptions (S1 Text), $N$ can be replaced, up to an unknown constant, by $1/f$. This yielded the formula for determining time $t$ in arbitrary units, over an interval of time which is unknown, but identifiable with an interval over inverse allele frequency $1/f$:

$$t_{a.u.} = \int_{(1/f)min}^{(1/f)max} \theta \cdot f \cdot d(1/f), \tag{1}$$

where $t_{a.u.}$ is the time in arbitrary units (a.u.), and $\theta$ corresponds to the exposure to clock-like mutational signatures. By computing $t_{a.u.}$ at all values of $1/f$, we obtained a vector of values $T_{a.u.}$ for the time in arbitrary units, from its minimum value, $\min(T_{a.u.})$ (0 by definition), to its

maximum value, $\max(T_{a.u.})$ (which corresponds to integration from $(1/f)_{min}$ to $(1/f)_{max}$. Confidence limits at 95% for time were calculated using the confidence bounds for mutational signature SBS1, taken as exposure $\theta$.

Time calibration in recurrent glioblastomas. In a subset of glioblastoma samples, we additionally know the time-to-relapse, denoted T, in months. We thereby identify the time course of the relapse [0; T] in months, with the time course in arbitrary units [0; $\max(T_{a.u.})$] (see the preceding paragraph "Time course" for the calculation of $T_{a.u.}$). To obtain $t$, the time in real units at any instant t in the interval [0; T], $t_{a.u.}$ is multiplied by the scaling factor T/$\max(T_{a.u.})$:

$$t = t_{a.u.} \cdot T/max(T_{a.u.}). \tag{2}$$

Time propagation from primary to recurrent glioblastomas. To obtain a link between the time course in the primary tumor, and the time course in the recurrent tumor, we studied the ratio of mutation accumulation between the end of primary tumor (subscripted 'P', taken as the last 5% time points) and start of recurrence (subscript 'R', first 5% time points), over the entire tumor. This is justified by the observation that the passage from the primary tumor to the recurrence corresponds to the instant of primary tumor resection. Using Eq 1 (S1 Text), this ratio could be written as follows:

$$\frac{(dM/dt)_P}{(dM/dt)_R} = \frac{(\mu(t) \cdot \pi \cdot \omega(t) \cdot \gamma(t) \cdot N(t))_P}{(\mu(t) \cdot \pi \cdot \omega(t) \cdot \gamma(t) \cdot N(t))_R}, \tag{3}$$

where subscript $i$ per subclone is not used, because we considered here that evolution parameters are taken over the whole tumor. The constant term $\pi$ can be normalized out of this ratio. Further, we have assumed that the mutation rate $\mu$ and division rate $\omega$ stay constant over this short period, because they are intrinsic features of the tumor cells. However, the count of tumor cells $N(t)$, and the tumor cell survival rate $\gamma(t)$ could not be considered constant. Regarding $N(t)$, we expressed it as the ratio of inverse allele frequency, since it is proportional to $N$ [5]:

$$\frac{N(t)_P}{N(t)_R} = \frac{(1/f)_P}{(1/f)_R}. \tag{4}$$

Of note, expression (4) is biased in practice by mutations which are not de novo in the recurrence, but inherited from the primary tumor. Ideally, only de novo mutations should be included to perform this calculation. Finally, the survival rate of tumor cells, $\gamma$, also could not be considered constant, and we had no indicator or surrogate for this value. For this reason, we have set an arbitrary value for the survival at end of primary tumor, relatively to the start of recurrence, $\gamma_P/\gamma_R = 1/300$.

Using the above, we could obtain expression (5) for $dM/dt$ at end of primary tumor:

$$\left(\frac{dM}{dt}\right)_P = \frac{(1/f)_P}{(1/f)_R} \cdot \frac{\gamma_P}{\gamma_R} \cdot \left(\frac{dM}{dt}\right)_R. \tag{5}$$

From this, and since the number of mutations at the end of the primary tumor, as well as the rest of parameters, was known, the time in real units at the end of primary tumor could be determined as follows:

$$dt_P = \frac{(dM)_P}{\frac{(1/f)_P}{(1/f)_R} \cdot \frac{\gamma_P}{\gamma_R} \cdot \left(\frac{dM}{dt}\right)_R}. \tag{6}$$

Similarly to the time course in the recurrence, the fact that the last instant of the recurrence, $dt_P$, was known, allowed to calibrate the time course $t$ in the primary at each instant, by multiplying by the scaling factor $dt_P/dt_{a.u.,P}$:

$$dt = dt_{a.u.} \cdot \frac{dt_P}{dt_{a.u.,P}}. \tag{7}$$

## Tumor cell survival ratio

For time calibration to real units, we have made an assumption on tumor cell survival ratio $\gamma_R/\gamma_P$ to determine real time in the primary tumor. To quantify the survival ratio, we have proceeded the other way around, using an assumption on the time course in the primary tumor, in order to determine the ratio $\gamma_R/\gamma_P$. We have utilized previously established values from [23] which estimates that the time between the most recent common ancestor (TMRCA) lies either 2 years or 7 years before primary tumor resection. As a consequence, we have assumed that these durations corresponded to the lowest and highest limits for the time course of the primary tumors The calculation of the tumor cell survival ratio was done by reformulating expression (6) into the following expression:

$$\frac{\gamma_R}{\gamma_P} = \frac{\left(\frac{dM}{dt}\right)_R}{\frac{(1/f)_R}{(1/f)_P} \cdot \left(\frac{dM}{dt}\right)_P}, \tag{8}$$

where all terms on the right hand side are known, either from the data ($dM$, $f$, $dt_R$) or from assumptions ($dt_P$).

## Tumor expansion profile

From Eq (8) in S1 Text, time and $1/f$ are proportional, with, as modulators, the growth rate $\omega$, the tumor cell survival rate $\gamma$, and the number of tumor cells $N$:

$$d(1/f) \propto \omega \cdot \gamma \cdot N \cdot dt, \tag{9}$$

where the increment $d(1/f)$ is known from the data, and increment $dt$ could be determined above, in section "Time course". As a consequence, one obtains the product $\omega \gamma N$, by dividing the increment $d(1/f)$ by the increment $dt$. The expansion parameters $\omega \gamma N$ are known within the constants of expression (8) from S1 Text, which are $\pi/K_{i,r}$. The curves giving $\omega \gamma N$ as a function of time are denoted expansion curves or expansion profiles.

## Curve segmentation

Segments for curves $M(1/f)$ were determined using R package segmented (v1.1–0), using an objective adjusted $R^2$ set to 0.995 (*P. virginalis*) and 0.9995 (GBM), and using the lowest number of segments which attained this objective, limited to a maximum of 20 breakpoints. *P*-values of significant changes between segments were evaluated using Davies' test (implementation from R package segmented). Because the regions before the first breakpoint, and after the last breakpoint, display marked and consistent differences with the general profile of the curve, we hypothesized that the automated segmentation revealed clonal mutations, or mutations which could originate from contamination by normal tissue (mutations with allele frequency lower than the first breakpoint), as well as mutations likely affected by the limit of detection of SNVs on sequencing data (mutations with allele frequency higher than the last breakpoint), We exclude those mutations, and restrict the accepted range to the interval between the first and last breakpoints.

## Classification of expansion profiles

Expansion profiles $\omega\gamma N(t)$ were subjected to curve clustering by k-means. To this aim, expansion profiles of 42 primary tumors were converted into a uniform arbitrary timeline of 1 to 1000, centered and standardized. Clusters were then determined using R package kml version 2.4.6, with default number of clusters and 5 redrawings. The quality of clustering was inspected using the criterion of Calinski-Harabatz (CH), which value was 25.43599 and 24.73392 for 3 and 4 clusters, respectively (CH values were in the 20.85733–22.32881 range for 2, 5 or 6 clusters). Because the partition into 4 clusters had a reasonably high CH value and was able to represent the "Increasing" type of curve while the partitions with 2 or 3 clusters did not, the partition into 4 clusters was retained.

## Statistical analyses

R [61] and Python [62] were used for all statistical analyses. Confidence intervals at 95% probability for the tree root in *P. virginalis* are taken as the 95% highest posterior density (HPD) interval. All statistical tests were unpaired and two-sided, with a level of significance set at 5%. Segmentation *p*-values correspond to the test of significant difference between segments (Davies' test, R package segmented). To check against overfitting, k-fold cross-validation of the segmentation procedure was conducted, with k = 10 folds. The resulting difference between test and validation mean square errors (MSE) was then examined. A non-increasing difference, when including one additional phase, indicates the absence of overfitting. Correlation coefficients between SBS1 and SBS5 were determined using Pearson method, and summarized by their median and IQR over GBM samples. A comparison between groups was made using an unpaired Wilcoxon rank-sum test. Differential time to recurrence between subgroups in the manually sorted $\omega\gamma N$ curves was assessed using a Kruskal-Wallis rank-sum test between curve types "Peak", "Increase" and "Paused Start". The "Convex" curve type was excluded, because there was only 1 instance of this kind of curve, in the subset of samples with infomation on the time to recurrence. The association of the $\gamma_R/\gamma_P$ ratio with the time to recurrence was assessed with a linear regression, using a simple or double $\log_{10}$ scale on the $\gamma_R/\gamma_P$ ratio, with Bonferroni adjustment.

## Supporting information

**S1 Fig. Mutation accumulation, selection and time dynamics of GBM tumors.** Panels A1-D1 show the primary tumor T1, panels A2-D2 show the recurrent tumor T2. (A) Mutation accumulation as a function of the inverse of allele frequency *1/f* (black) and phases from automated segmentation (breakpoints (grey) and segments (blue)). The confidence band at 95% level is indicated in grey. (B) Nonsynonymous to synonymous ratio. Purple and blue stars show nonsynonymous and synonymous mutations, respectively. The smoothened ratio is shown in red. (C) Clock-like and non-clock-like mutational signatures. (D) Mutation accumulation as a function of time.
(PDF)

**S2 Fig. Transition of primary tumor to recurrent tumor.** (A) Dynamics of growth rate $\omega$ times tumor cell survival rate $\gamma$ times number of cells *N*, for (P) the primary tumor and (R) the recurrence. (B) Time-resolved mutation accumulation for primary tumor and recurrence.
(PDF)

**S1 Text. Supplementary Methods.**
(PDF)

**S1 Table. Supplementary Table 1.** *P*-values for test of difference between segments for P. virginalis and glioblastoma samples.
(DOCX)

**S2 Table. Supplementary Table 2.** Characteristics of tumor cell survival ratio $\gamma_R/\gamma_P$ (n = 20).
(DOCX)

**S3 Table. Supplementary Table 3.** *Procambarus virginalis* Samples.
(DOCX)

**S4 Table. Supplementary Table 4.** Raw sequencing data for new and resequenced Procambarus virginalis samples.
(DOCX)

**S1 Data. Source Data for Figures.**
(XLSX)

## Acknowledgments

We would like to thank Verena Körber and Thomas Höfer for helpful discussions and for providing data, and Julian Gutekunst for discussions about the methods. We would also like to thank Katharina Hanna for data and for crayfish culture, and Sina Tönges for sample processing. We further acknowledge the German Cancer Research Center Genomics and Proteomics Core Facility for whole-genome sequencing.

## Author Contributions

**Conceptualization:** Carine Legrand, Ranja Andriantsoa, Frank Lyko.

**Formal analysis:** Carine Legrand.

**Methodology:** Carine Legrand.

**Resources:** Peter Lichter.

**Supervision:** Frank Lyko.

**Writing – original draft:** Carine Legrand.

**Writing – review & editing:** Carine Legrand, Günter Raddatz, Frank Lyko.

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
