## [Decision Letter · Decision Letter 0]

26 Jun 2023

Dear Dr Legrand,

Thank you very much for submitting your Research Article entitled 'Time-resolved, integrated analysis of clonally evolving genomes' to PLOS Genetics.

The manuscript was fully evaluated at the editorial level and by independent peer reviewers. The reviewers appreciated the attention to an important problem, but raised some substantial concerns about the current manuscript. Based on the reviews, we will not be able to accept this version of the manuscript, but we would be willing to review a much-revised version. We cannot, of course, promise publication at that time.

If you decide to revise the manuscript for further consideration at PLOS Genetics, please aim to resubmit within the next 60 days, unless it will take extra time to address the concerns of the reviewers, in which case we would appreciate an expected resubmission date by email to plosgenetics@plos.org.

We are sorry that we cannot be more positive about your manuscript at this stage. Please do not hesitate to contact us if you have any concerns or questions.

Yours sincerely,

Ville Mustonen

Guest Editor

PLOS Genetics

Justin Fay

Section Editor

PLOS Genetics

Reviewer's Responses to Questions

**Comments to the Authors:**

Reviewer #1: In this manuscript, Legrand and colleagues develop a framework to study mutation accumulation in asexual organisms, in which they look at the relationship between mutation accumulation and variant allele frequency as a proxy for mutational age. This allows disentangling the links between mutation rate, tissue growth and survival of de novo mutations. Authors apply their framework in both a seemingly young parthenogenetic species (the marbled crayfish Procambarus virginalis) and clonally-evolving tissues (glioblastoma), analyzing previously published data and some newly sequenced crayfish sequences.

My background is in empirical population genomics and I have some experience working with non-model parthenogenetic species. My expertise is very limited regarding cancer evolution. As a result, my comments below reflect my expertise (or lack thereof) on these different aspects. Overall, I enjoyed reading the paper, from what I can tell the approach is original and mostly sound and the analyses are rigorous (but see my first major comment below). I however believe that the authors could do a better job at demonstrating the importance of their findings and its broad interest to the PLOS Genetics audience (see major comment 2).

Major comment 1 – The framework heavily relies on a precise estimation of the mutation rate, which is a complex endeavor for non-model organisms. Thankfully, authors can rely on a marbled crayfish line which was maintained in the lab over seven years (2012-2019), basically providing a mutation accumulation experiment. Authors used whole-genome sequencing of three individuals sampled in 2012, 2017 and 2018 to call SNPs using state-of-the-art methods.

1.1 – First, it is not possible from this manuscript alone to assess the quality of the resequencing data, and I believe a reference was omitted L294-295 (“These new samples were prepared and submitted for whole genome sequencing following the protocol already described”). From Table S3 in Gutekunst et al. 2018 (which data were partly re-analyzed for this study), the average sequencing depth for the 2012 sample (“Animal 1”) seems to be around 72x, but no information is provided in the current manuscript regarding the number of reads per individual, the sequencing strategy (I guess 2x150 bp) or the average sequencing depth in the 2017 and 2018 samples, while deep sequencing is crucial for accurate SNP calling in the context of de novo mutation detection. The reader can just assume sequencing depth is high, since “variants with quality≥35, coverage≥50 and ≤200 were retained for the main estimate of the mutation rate” (L314). More details on the data would help readers assess their quality.

1.2 – My second and more important point stems from this sentence L314, as authors identified mutations at sites where depth was comprised between 50 and 200 – what would be defined as callable sites. After finding candidate de novo mutations in the two more recent individuals (how many per individual? Where they at heterozygous or homozygous states?), they divided their count by the total count of nucleotides in the triploid genome (10.5 Gb, L320). Unless authors made a typo, this is not correct: the denominator should not be the total genome size, but the number of callable sites (i.e., the fraction of the genome that could actually be surveyed). For a correct estimation of the mutation rate, authors could for example look at Lopez-Cortegano et al. (doi:10.1093/molbev/msab140, see p. 3719). I would assume this error biases downwards the mutation rate estimated for marbled crayfish, which may have implications for the calibration of the tree inferred with BEAST (Fig. 1C) and the history of the parthenogenetic lineage as a whole (e.g., L76, L269-271). Note that a similar error might have occurred with the dN/dS ratio analysis, but it is unclear in the current version how the “average quotient” was estimated for the full genome (L366). I would kindly ask the authors to estimate the mutation rate for marbled crayfish with a correct method (callable sites can easily be identified from BAM files using mosdepth, doi: 10.1093/bioinformatics/btx699), and update their findings accordingly.

Major comment 2 – While I usually enjoy short papers, I believe this manuscript would benefit from a more detailed writing in many places. As it combines theory and empirical data analysis in both a non-model organism and tumors, additional details would help convey the context, importance and broad interest of these findings. Below I provide some examples of parts where I would want to get more detailed information, trying to rank these by perceived importance.

2.1 – Introduction: I would expect some framing of the scientific question before getting into the specifics of the study systems. At the moment, it is unclear what is gained by developing “an integrated analysis of clonal genome evolution”, but it should become apparent in the first few paragraphs of the introduction. Likewise, the marbled crayfish is a puzzling system, and more details should be provided in the section L68-77: genetic differentiation compared to what (L74)? What is the “particular mode of asexual reproduction” (L71)? What is the specific event mentioned L76? Is parthenogenesis obligatory (are we sure there are no unintentional crosses in the lab)? More information on the origin of this species would help understand its value for the current study.

2.2 – Discussion: After reading the manuscript several times, I am still not sure which are the “important insights” (L286) being delivered. What does the framework tell us about clonal evolution, and/or the evolution of glioblastoma? It would help if authors drew more connections to relevant literature. To provide an example (but as stated above, I am not an expert in cancer genomics), are the pervasive signatures of selection detected in some samples (L206) in line with recent findings (e.g., Tilk et al. 2022, doi: https://doi.org/10.7554/eLife.67790)? What is the general impact of selection on the results? What is the (clinical? biological?) relevance of the four subtypes detected (L276)? Finally, which are the potential limits of this work and what could be added?

2.3 – Results: The rationale explained in the S1 file should be integrated a bit more in the main text, as the approach is currently quite vague (L133-135).

I cannot interpret the comparison of clock-like and non-clock-like mutational signatures as it stands, and I don’t get why human mutational signatures are used for crayfish genomes (I must be missing something). Are expansion parameters (L231) loosely linked to any other measure used in cancer genomics? The invasion history of the marbled crayfish should be more detailed on L120-127.

Below are some minor comments:

L57-58: in my understanding, the term “mutation” is used for two different things here, the process itself L58 (the apparition of a new variant in the pool), and its consequence L57 (the new variant itself, i.e., a substitution). Maybe consider editing.

L92: unclear why “time” here.

L96-100: see major comment 1, but please add more information here (number of individuals, average depth…).

L105: please provide references for known mutation rates in the main text on top of the legend of Fig. 1. Also it seems there is at least one mutation rate estimated for Crustaceans (https://doi.org/10.1186/s12862-021-01836-3).

L109: grey colors are not displayed on the Fig. 1.

L130: is there any outgroup available to root the tree? If not, is it a problem?

L144: is the dN/dS ratio actually computed from fixed differences, or from polymorphisms (so pN/pS)?

L150: red instead of blue.

L153: grey instead of red.

L167-168: from what I understood, the dating of the marbled crayfish lineage comes from the (incorrect) estimation of the mutation rate, not the framework, which seems at odds with the statement L91-93.

L174: “better understanding of *its* evolutionary parameters”.

L178-179: Fig. 3A not 4A.

L204-212: this heterogeneity is interesting and could be discussed further.

L244: is this only for the tumor shown in Fig. 3, or for all of them? Likewise, what is the impact of the assumption made on the TMRCA L444-445? Please clarify.

L262-264: this statement seems at odds with L163-165 and L266-269. More generally, this makes me wonder if I really understand how the framework disentangles between variable mutation rate and growth. This could be made clearer in the main text.

L295: please add references for the sequencing protocol etc…

L302-305: the reference genome was built from short reads and is quite fragmented, is this supposed to have an impact on mutation detection? Likewise, was anything done to remove sites close to indels?

L348: define “features of the subclone”.

L351: is this assumption of constant ploidy reasonable for tumors? Please elaborate.

L371: maybe introduce a bit what are these mutation signatures.

L383: are these good indicators for time in the crayfish?

L477-479: as is, I don’t get the inherent value of this classification.

The data availability section only mentions already published sequencing data (both for the marbled crayfish and glioblastoma). There is no mention of samples re-sequenced for this study (L290: samples Madagascar 1 and Moosweiher, L292: animal 34 and animal 35). The numerical data that underlies figures is not provided. Finally, and while it is not stated in PLOS' data availability policy, scripts are also not provided, but I think these should be available somewhere, for reproducibility and if readers are interested in your framework’s implementation.

Reviewer #2: Legrand et al. apply mutational timing methods previously introduced in the cancer genomics literature to infer evolutionary parameters from two types of clonally evolving populations: one cohort of clonally evolving marbled crayfish including both published and newly generated datasets, and one published dataset of glioblastoma patients. The authors should be commended for the integrative analysis of clonal genomes in both species and cancer evolution. That being said, it is important that for this paper to be accepted, the authors address some major technical concerns summarized below.

Major points:

- The authors did not account for potential confounders of the background mutation rate of each gene in their dN/dS analyses. They should control for the sequence composition of the gene and mutational signatures, by e.g. using a method like dNdScv that avoids common mutation biases affecting dN/dS using trinucleotide context-dependent substitution matrices.

- The automated segmentation of inverse allele frequency spectra M(1/f) appears to overfit the data:

> Examples of this are statements like: “In the exemplary sample 1 in Fig 3A, the segmentation separates 5 phases significantly.”

> The authors should reexamine the frequency spectrum using previously published methods to segment the spectra, by distinguishing mutations in the non-clonal tail from clonal and subclonal mutations (e.g. using SNV clustering methods like MOBSTER).

- In the primary-recurrent analysis, the classification of expansion profiles is extremely weak. Expansion profiles ωγN(t) are classified visually and manually curated without further explanation. This entire analysis should be removed unless a well-grounded methodology can be used to define expansion profiles (e.g. using unsupervised clustering).

- Could the authors expand on the impact of ploidy on their analysis? Particularly in the case of GBM, where the tumors often have a substantial number of copy number aberrations. They should ensure to limit their timing analysis to diploid regions. Could they also speculate in the Discussion about combining their approach with the timing of copy number gains using SNVs in amplified regions?

- Could the authors demonstrate that a lower coverage threshold >15 is justified for coalescent time analyses? Are noisy variants affecting the coalescence estimates?

Minor points:

- In line 183, could the authors explain why they expect that mutation frequencies in the high-frequency phase (phase 1) are expected to include artifacts? This should not really be an issue unless germline variants appear to be clonal in the GBM samples and have not been exhaustively filtered out.

- In lines 178 and 179, the figure callouts should refer to Fig. 3 instead of Fig. 4.

**Have all data underlying the figures and results presented in the manuscript been provided?**

Reviewer #1: **No: **The data availability section only mentions already published sequencing data (both for the marbled crayfish and glioblastoma). There is no mention of samples resequenced for this study (L290: samples Madagascar 1 and Moosweiher, L292: animal 34 and animal 35).

The numerical data that underlies figures is not provided.

Finally, and while it is not stated in PLOS' data availability policy, scripts are also not provided.

Reviewer #2: Yes

PLOS authors have the option to publish the peer review history of their article (what does this mean?). If published, this will include your full peer review and any attached files.

Reviewer #1: No

Reviewer #2: No

---

## [Decision Letter · Decision Letter 1]

27 Nov 2023

Dear Dr Legrand,

We are pleased to inform you that your manuscript entitled "Time-resolved, integrated analysis of clonally evolving genomes" has been editorially accepted for publication in PLOS Genetics. Congratulations!

Yours sincerely,

Ville Mustonen

Guest Editor

PLOS Genetics

Justin Fay

Section Editor

PLOS Genetics

Comments from the reviewers (if applicable):

Reviewer's Responses to Questions

**Comments to the Authors:**

Reviewer #1: I believe authors have addressed all my comments appropriately and, in my opinion, those of the other reviewer as well. I am looking forward to seeing this in print.

Reviewer #2: The authors have address most of my comments or described them as limitations of the data in the Discussion.

**Have all data underlying the figures and results presented in the manuscript been provided?**

Reviewer #1: Yes

Reviewer #2: Yes

PLOS authors have the option to publish the peer review history of their article (what does this mean?). If published, this will include your full peer review and any attached files.

Reviewer #1: No

Reviewer #2: No

**Data Deposition**

http://datadryad.org/submit?journalID=pgenetics&manu=PGENETICS-D-23-00320R1

**Press Queries**

---

## [Editor Report · Acceptance letter]

8 Dec 2023

PGENETICS-D-23-00320R1 

Time-resolved, integrated analysis of clonally evolving genomes 

Dear Dr Legrand, 

We are pleased to inform you that your manuscript entitled "Time-resolved, integrated analysis of clonally evolving genomes" has been formally accepted for publication in PLOS Genetics! Your manuscript is now with our production department and you will be notified of the publication date in due course.

With kind regards,

Zsofi Zombor

PLOS Genetics

On behalf of:
